# Measuring the Impact of Bedroom Privacy on Social Networks in a Long-Term Care Facility for Hong Kong Older Adults: A Spatio-Social Network Analysis Approach

**DOI:** 10.3390/ijerph20085494

**Published:** 2023-04-13

**Authors:** Aria C. H. Yang, Habib Chaudhury, Jeffrey C. F. Ho, Newman Lau

**Affiliations:** 1School of Design, The Hong Kong Polytechnic University, Hong Kong SAR 999077, China; 2Department of Gerontology, Simon Fraser University, 8888 University Drive Burnaby, Vancouver, BC V5A 1S6, Canada

**Keywords:** long-term care home, older adults, physical environment, social network analysis, compact living

## Abstract

This study aims to measure the impact of bedroom privacy on residents’ social networks in a long-term care (LTC) facility for older adults. Little is known about how the architectural design of bedrooms affects residents’ social networks in compact LTC facilities. Five design factors affecting privacy were examined: bedroom occupancy, visual privacy, visibility, bedroom adjacency, and transitional space. We present a spatio-social network analysis approach to analyse the social network structures of 48 residents. Results show that residents with the highest bedroom privacy had comparatively smaller yet stronger groups of network partners in their own bedrooms. Further, residents who lived along short corridors interacted frequently with non-roommates in one another’s bedrooms. In contrast, residents who had the least privacy had relatively diverse network partners, however, with weak social ties. Clustering analyses also identified five distinct social clusters among residents of different bedrooms, ranging from diverse to restricted. Multiple regressions showed that these architectural factors are significantly associated with residents’ network structures. The findings have methodological implications for the study of physical environment and social networks which are useful for LTC service providers. We argue that our findings could inform current policies to develop LTC facilities aimed at improving residents’ well-being.

## 1. Introduction

The impact of the physical environment in long-term care (LTC) facilities on residents’ social relationships is increasingly recognized [1,2,3]. In particular, the design of the resident bedroom is believed to be one of the most crucial aspects in LTC facilities [4,5,6]. Previous studies have found that privacy in resident bedrooms has a significant influence on residents’ social interaction patterns [7,8]. Although privacy is a fundamental human need [9], it is often overlooked in institutional settings and especially scarce in compact LTC facilities [10]. The lack of privacy in bedrooms has a negative impact on residents’ social relationships [7,8,11], which is critical to cognitive functions and quality of life [12]. However, most of the studies in environmental gerontology focus on the impact of the physical on behavioural outcomes and little is known about its effects on social relationships. Moreover, these studies were mostly conducted in the Western context where the population density and housing typologies are different from high-density Asian cities. While there is an increasing focus on privacy in resident bedrooms in Hong Kong LTC facilities [13,14], the majority of studies have only investigated privacy and social relationships qualitatively rather than quantitatively. This study is necessary because it utilises indoor location-tracking technologies which raises the possibilities to examine the relationships between the privacy of resident bedrooms and social networks. To the best of our knowledge, this study is the first research attempted at addressing the knowledge gaps in the design of LTC facilities. Social network analysis is increasingly used as a creative approach to evaluate and assist in the understanding of social relationships that are not readily available through conventional methods in healthcare research [15]. The approach captures the social relationships of residents from a sociocentric or whole-network perspective, rather than person-centred networks. In addition, within the limited studies on social relationships in Asian LTC facilities, most of them have explored social relationships from an individual-centred perspective [13,16], as opposed to a whole network perspective [17,18]. The sociocentric technique has some advantages over personal network research, including better measurement of dyadic (person-to-person) social interaction and knowledge of the general network structure of the local social network. However, routine daily activities in LTC facilities, such as dining, resting, and chatting do take place in the physical environment. The increasing use of ubiquitous location tracking technologies is useful for capturing the dyadic social interaction in specific locations inside the LTC facilities. The main intellectual goal of our research is therefore to figure out how to integrate and use a spatial and social network analysis approach to investigate everyday social life in LTC facilities, focusing especially on social networks.

In this study, we will present the use of an integrated spatio-social network analysis approach to assess the social network structure between residents of a Hong Kong LTC facility. Specifically, this study has two objectives: (1) to identify residents’ location-based social interaction patterns and social network structure via analysed ubiquitous location data, and (2) to determine the associations between bedroom privacy and social network structure.

### 1.1. Bedroom Privacy

Privacy is defined as “selective control of access to the self or to one’s group” [9]. The provision of privacy has a direct impact on the amount of socialisation; too much privacy could lead to social isolation and too little privacy may result in unwanted intrusion [9]. Privacy can be regulated through the control of spatial mechanisms in terms of primary, secondary, and public territories [9].

Privacy in the primary territory refers to the amount of personal space that the residents have access to in the bedrooms. Previous study has shown that lower bedroom occupancy enables residents to have more control over privacy which gives them a sense of security and ownership [19]. Compact facilities in Hong Kong are characterised by a large unit size and high occupancy rate in multiple-bed rooms [14]. Personal space is frequently compromised with unintentional intrusions from other roommates or staff who enter the bedroom to give care services [20,21,22]. However, another study suggests that a balance needs to be achieved as many Chinese patients in nursing units preferred to be seen by the staff from nursing stations out of concerns for safety [23]. 

The secondary territory refers to the transitional area between the resident bedrooms and common areas. Previous research suggests that having a full range of spaces from private, to semi-private, to semi-public, and to public spaces is vital for residents’ social relationships [24,25,26]. The presence of transitional areas has been shown to encourage socialisation in an informal setting [1,27,28]. Comfortable seating along the corridors would encourage meaningful social interactions [1]. The proximity between the bedroom and transitional space such as patios and alcoves near the bedrooms can help to regulate the perception of privacy and social interaction [27,28]. 

Public territory refers to the common areas, such as lounge. Having good visual access between the bedroom and common areas preserves residents’ dignity and privacy and allows residents and staff to see and be seen by one another [29]. Being able to see what is happening in and around the common areas enables residents to make decisions about when and where to go [29,30,31]. Particularly, being able to see the dining room from the bedrooms increases the opportunity for socialisation [29,32]. Allowing residents to be seen by the staff is an important way to facilitate communication and interaction [29,33]. Nevertheless, it is also worthwhile to consider flexible measures when the residents do not wish to be seen [4,31]. 

Summarising previous studies shown above, five architectural design factors were identified when considering bedroom privacy in compact LTC facilities: bedroom occupancy, presence of transitional spaces, bedroom adjacency, visibility (to see), and visual privacy (to be seen).

### 1.2. Social Life in LTC Facilities

The body of research on the value of social connections in long-term care is expanding. There are generally two lines of social support: structural support (for example, network size) and functional support (for example, availability of networks). Among residents living in LTC facilities, perceived social support, a cohesive environment, social activities, and family contact have all been connected to resident well-being and quality of life [34,35]. It has become increasingly evident that resident social life is closely associated with the design of the physical environment [2,3,32,36].

The capacity of social network analysis (SNA) in determining number of network partners and social ties has advanced the comprehension of social life among residents [15,18,37]. Social life can be measured with social network characteristics such as size and frequency. The work by Abbott and colleagues [17] shows that there is a low social integration among residents in an assisted living facility. In Hong Kong, residents were found to have between 0.6 and 2.6 perceived social networks which suggests a prevalence of social isolation [13,16]. In the West, previous studies have found that residents with Alzheimer’s disease had noticeably less perceived social networks compared with those without [18,38]. Another study developed SNA as a framework to detect older adults in palliative care who were at risk of social isolation [37]. The authors tracked the social interactions between older adults in palliative care, and formal and informal caregivers to detect social isolation among older adults using SNA. Such knowledge would serve as a potential indicator of older adults’ psychological outcomes and cognitive functioning to better guide dementia care [38,39,40,41]. Abbott and colleagues [17] utilised network visualisation or sociograms in SNA to show the number of connections as well as the degree of social integration between residents and staff. SNA could advance LTC service providers’ understanding of residents’ expectations and perceptions of social life in LTC facilities [18]. The study of social networks among older adults has mainly focused on older adults who are living in the community with relatively good health [42,43], and little is known about the social networks of older adults who have considerable functional decline and entered LTC facilities. The social clusters among residents in LTC facilities is under-studied.

Having different types of social network partners, for example, social connections with other residents and staff in LTC facilities, is one of the most significant determinants of residents’ well-being and life satisfaction [40,41]. Each social relationship, including peers, family, and staff, has a specific contribution to residents’ well-being [39]. Residents in Australia reported that friendship is an important type of social network [10]. Early research in this field were conducted by Wenger [44] who classified individuals’ networks into five clusters: locally integrated, wider community-focused, local self-contained, family dependent, and private restricted. Based on Wenger’s social network types, researchers have further developed four key network types among older adults: diverse cluster, family-focused cluster, friend-focused cluster, and restricted cluster [42,45,46]. A diverse cluster is similar to Wegner’s locally integrated cluster which is characterised by active involvement with family, friends, and neighbours. A family-focused cluster is about relationships that are reliant on family members (family dependent). A friend-focused cluster depicts relationships with friends instead of family or neighbours (wider community-focused). A restricted cluster describes a low level of interactions with anyone which is considered the same as private restricted.

### 1.3. A Spatio-Social Network Analysis Approach

Given the spatial qualities of residents’ social life described above, how can we measure and evaluate the design of bedroom privacy in compact LTC facilities? Since most of the previous studies on social networks have not examined the interplay between the physical and social constructs, we plan to answer this main research question by making use of the rapidly growing body of SNA literature. 

The role of physical space on the social networks in the LTC facilities cannot be ignored, especially when residents spend much of their day dining, sitting, and sleeping in a compact and confined environment. Although SNA is not solely concerned with accounting for physical location, it is increasingly being adopted in this way to study the effects of morphological qualities in the built environment on social networks [47]. There is a growing body of research on the study of location- and activity-based (two-mode) social networks in the design and planning of cities. Integrated spatio-social SNA has been used to demonstrate segregation between migrants and local residents in China [48]. The methodological approach employed in our study is directly based on this significant recent advancement in the academic field of SNA.

Figure 1 shows two different kinds of networks. Consider an LTC facility with a limited number of physical locations (such as resident rooms, common areas, etc.), each accommodating a certain activity or event that attracts a given group of residents. Since each location also occupies a specific privacy condition (private, semi-private or public), a location-and-user (two-mode) activity network can be visualised to show the users’ visit to specific locations. This two-mode network can then be further converted into a user-and-user (one-mode) social network, illustrated as shown in the bottom-left corner of Figure 1. A social network is made up of two elements: the network members (nodes) and the interaction ties (edges). These elements are first calculated within the two-mode network where residents’ entire social network clusters and types are identified according to locations. Following that, social networks are then represented by the key characteristics of network structure which include network size, number of social ties, density, and centrality.

## 2. Materials and Methods

### 2.1. Settings

A spatio-social network approach is especially suited for LTC facilities since networks within tend to be closed, and the requirements of collecting data from a bounded group can be met. By distinguishing ties that originate from within those LTC facilities, a natural network boundary can be defined [49]. This study was conducted in an LTC facility in Sham Shui Po district of Hong Kong. The facility is a six-storey building that accommodates 200 residents. The ground floor consists of a reception area and a large multi-functional room. Each floor contained a typical residential unit for 40 residents which consists of a centralised nursing station and four types of shared bedrooms: four-bedrooms facing common areas (common); four-bedrooms facing a corridor (corridor); three-bedrooms, and five-bedrooms (Figure 2); 80% of the residents lived in four-bedrooms, 8% lived in three-bedrooms, and 12% lived in five-bedrooms. Residents from different floors and room types were recruited. The staff members did not wish to participate in the study.

### 2.2. Assessing Bedroom Privacy

The overall privacy score according to Sheffield Care Environment Assessment Matrix (SCEAM) [31] was rated at 0.38 in each unit, which indicates a rather low level of spatial privacy. However, the SCEAM does not allow the evaluation of privacy on the bedroom level. The specific variables used to assess bedroom privacy vary considerably across studies. For example, the Therapeutic Environment Screening Survey for Nursing Homes (TESS-NH) [31] considers privacy through the aspects of partitions and sound. On the other hand, SCEAM emphasises on the visual and acoustic privacy during care activities and residents’ access to private telephone conversations. The Environmental Audit Tool (EAT) [29] discusses privacy and social interaction through the concept of seeing and being seen, such as the ability to see the dining room from the bedroom and the visual access between the different common areas. 

Guided by Altman’s [9] model of privacy based on territoriality, the authors have developed a specific scale and geometrical indicators established to determine the quality of privacy in shared bedrooms (Table 1). The higher the total score is, the more privacy the bedroom has to offer. Bedroom occupancy considers the number of residents who live in the bedroom which ranges from the preferred number of one to two residents [24,50] to three, four, and five residents as found in this study. Bedroom adjacency concerns the order and continuum of spaces, ranging from private, to semi-private, to semi-public, to public [25,26]. Transitional space concerns the number of transitional spaces between the bedroom and common areas [1,27,28]. Visibility focuses on the aspects of being able to see other residents and staff from the bedroom [29,30]. Visual privacy addresses residents’ control over being seen by others in the public space [23,31].

### 2.3. Measuring Social Networks

Having active social networks in close geographical proximity plays a significant role in the health outcomes of the general population [51]. In LTC facilities, these social networks are the residents who live together. In our study, residents’ locations around their own bedrooms, other residents’ rooms, and the common areas were tracked ubiquitously by a Bluetooth Low Energy (BLE) indoor positioning system for a one-month period. The locations in the entire building were used to study the closed networks. The location data recorded residents’ day-to-day interactions with their social network partners, i.e., roommates and non-roommates. The data were analysed into user-location (two-mode) networks and then converted into user-user (one-mode) interaction where facility-level socio-centric networks can be constructed to visualise the social network of all participating residents (Figure 1). 

Collecting network data in a closed environment such as an LTC facility unit is less prone to measurement errors, because ideally, all physically and socially active members of the unit are recruited, and each interaction is tracked on the BLE indoor positioning system. This is important for accuracy because of the possibility for residents to underreport or overstate social interactions when conducted manually which may misrepresent the extent of their social networks [52]. The sociocentric technique has some advantages over egocentric or personal network research, including better measurement of dyadic (person-to-person) social interaction and knowledge of the overall network structure of the local social network [53]. This is especially true for relationships involving residents with cognitive impairments which is found in most residents in LTC facilities. Since the best methods to obtain social network information of residents on location-tracking technologies have yet to be identified, we also collected data continuously for 24 h to observe network patterns outside of institutional daily schedules.

### 2.4. Data Collection

In total, 50 residents were recruited to participate in the study. Consent to participation was obtained from the family members if the residents had cognitive impairment. Purposive sampling was deployed for the recruitment. The inclusion criteria were those who were aged 65 and over and able to move about in the facility independently. The sampling size of each bedroom type aimed to resemble the distribution on a facility level. Table 2 shows that of the 48 participants, 7 lived in three-bedrooms (14.6%), 18 in the four-bedrooms facing corridors (37.5%), 15 in four-bedrooms facing common areas (31.3%), and 8 in five-bedrooms (16.7%). One face-to-face training workshop was held to introduce the study to the participants. Descriptions of the study and an informational leaflet were given. After that, private sessions were arranged with each participant to address any concerns over the participation. The participants were given the opportunity to ask questions, think things over, and consult with a friend or family member if necessary. Residents who were interested provided their informed consent.

Of the participating residents, 48 residents (96%) continued to wear a smartwatch throughout the study. The two remaining residents declined to wear the smartwatch and their data were subsequently omitted from the dataset due to missing data. According to the space-use records, the BLE indoor location-tracking system showed an average accuracy of 85% in detecting resident locations. The said precision is compromised by the fluctuation of Wi-Fi signals in the facility, and reflection, diffraction, and transmission loss around people, objects, and structures which resulted in lost and/or duplicated data on multiple sensors [54]. In our study, it took two months to adjust the positioning of the sensors to minimise the interference through a series of trials and tests. The official location-tracking resulted in a dataset of 139,276,800 second-by-second location data entries. The dataset was then aggregated into 18,200 hour-by-hour location records over the course of one day. The collection of location data about the resident’s social interactions with others took a sociocentric approach. Each resident wore a smartwatch that had its distinct MAC addresses. Enabled by BLE technology, a Raspberry Pi (RPi) placed in each bedroom, pantry, and common area continuously scanned for the presence of individual smartwatches. Given the limitations of the system storage, the location data were collected at alternating hours for the entire 24 h each day. The smartwatches needed to be charged once every week which was carried out by the staff on the evening shift. A typical engagement lasts one minute and thirteen seconds, with only 5% of recorded interactions lasting longer than five minutes, according to Montanari and colleagues [55]. Therefore, when two or more smartwatches were detected to be in the same room or space for more than 5 min, it is assumed that a meaningful social interaction happened in that room or space. 

### 2.5. Analysis

Objective 1 was to identify residents’ location-based social interaction patterns and social network structure. To meet the first part of the objective, ubiquitous location data were calculated to obtain two-mode user-to-location networks in Microsoft Excel. Following that, user-to-user networks based on locations were then developed manually in Excel to identify sociocentric interaction patterns. In this study, social networks were measured between residents who were roommates and non-roommates. The three sets of variables included were number of social network partners (roommates and non-roommates) in the bedrooms or common areas, frequency of contact with social network members in bedrooms or common areas, and degree of centrality. The second part of objective 1 was met by using social network analysis software Ucinet Version 6 [56] which provided the details about residents’ network structures, i.e., nodes, edges, social ties, and centrality for network visualisation. Based on the social network structure, we further categorised individuals into social clusters using non-hierarchical k-means cluster analysis in SPSS to identify the different social cluster types. Based on the calculation of their Euclidean distances from cluster centres, cluster analysis uses a stepwise strategy to identify groups of individuals that are homogenous among themselves but as dissimilar from other groups of persons as feasible. K values used varied from 1 to 10. To determine the optimal value of k, the Elbow method, a bend in the plot, is generally considered as an indicator of the appropriate number of clusters. To make sure that the clusters can be interpreted meaningfully, cluster results must also be examined conceptually. 

Objective 2, to determine the associations between bedroom privacy and social network structure, was met by conducting a multiple regression analysis between network characteristics and privacy assessment results. The influence of bedroom privacy was addressed in this analysis to predict network characteristics from bedroom privacy factors, in which the social network structure variable is predicted based on the value of the bedroom privacy. 

## 3. Results

### 3.1. Bedroom Privacy

Two researchers evaluated the quality of privacy in the four different bedroom types according to the five design factors (Table 1). Results show that three-bedrooms had the highest privacy score, followed by four-bedrooms (corridor), five-bedrooms, and four-bedrooms (common). Table 3 illustrates the privacy score of each architectural feature in each bedroom. Although the five-bedrooms accommodated the high number of residents, it was rated having privacy than the four-bedrooms (common) which lacked any visual and sound buffers from the common areas. Observations on site showed that the residents of this room type hung clothing and towels around their bed spaces to maintain their personal space.

### 3.2. Feasibility of the Spatio-Social Network Analysis Approach

In the context of this paper, instead of illustrating the user-user (one-mode) network, the user-location (two-mode) networks were first visualised. Two-mode social networks were used to represent the networks between two types of entities. The nodes therefore represent participants and locations visited, and the meaningful interactions represent the edges. Based on the information of the number of network partners, it was also possible to visualise a user-location social network to illustrate the residents’ space-use patterns at the different locations (Figure 3). The most used spaces were four-bedrooms facing common areas and own bedrooms. The least visited spaces were three-bed, five-bedrooms, and the lobby.

### 3.3. Residents’ Social Networks

#### 3.3.1. Social Network Structure

Each participants’ social network partner (node), number of social network partners (edge), frequency of social interaction (tie strength), and degree centrality (number of links held by each node) were calculated individually according to bedroom types (Table 4). The frequency of contact participants had with their network partners was calculated based on the number of times social interactions were registered. 

Figure 4 illustrates the social network in the LTC facility. While there are no standard protocols for drawing these networks, generally nodes with greater centrality are placed in the centre of the drawing and less central nodes are placed along the periphery. There are people who are central to the networks and others who are on the periphery, with one person being fairly isolated (P05). Since the indoor location tracking system did not track in-coming or out-going interactions, we do not have the measures for in-degree and out-degree ties. Among the participants of three-bedrooms (circle nodes), a large proportion (86%) belonged to two distinct social clusters. They have an average of 1.10 social network partners. The frequency of interaction with others is 2.86 times in a day. Their mean degree centrality to the network is 0.07. The participants of four-bedrooms facing common areas (square nodes) are typically in the centre of the social clusters. They have an average of 2.43 social network partners with a mean interaction frequency of 4.60 times per day. Nevertheless, they scored the lowest degree centrality of 0.04, which indicates weak social ties with others. The participants of four-bedrooms facing a corridor (triangle nodes) have well-distributed social partners in their networks. They have an average of 1.85 social network partners with a mean interaction frequency of 4.44 times per day. They have strong social ties with others. These participants have the highest degree centrality of 0.23 which suggests that they play an important role in holding the network together. The participants of five-bedrooms (diamond nodes) are at the periphery of the network. They have an average of 2.79 social network partners and a interaction frequency of 2.75 times. The degree centrality for this group is 0.06 which suggests a relatively low importance in the overall network.

#### 3.3.2. Social Network Types

Based on the k-means clustering analysis, we identified five social clusters within the entire network (Table 5). This conceptual and statistical strategy worked together to help us decide on the five-cluster option. We display the means for the factors that were used to create the five groups in Table 5. It is helpful to consider the factors that differ most between groups when analysing the groupings and giving them a meaningful cluster label. In addition to reflecting a continuum from more socially integrated to more socially isolated, groups are listed in order of prevalence. Ten per cent of the total number of participants belonged to Cluster 1: Diverse (common area) social cluster which was characterised by the highest mean number of non-roommates as network partners in others’ bedrooms and the highest frequency of interaction with roommates and non-roommates in the common area. Twenty-one per cent of the participants belonged to Cluster 2: Diverse (bedroom) cluster which was characterised by the highest mean number of roommates in the bedroom area, as well as the highest frequency of interaction with roommates and non-roommates in the bedroom areas. This social cluster was also highlighted by participants who had the highest frequency of being alone in common areas after midnight. Twenty-three per cent of the participants belonged to Cluster 3: Non-roommate-focused social cluster. This cluster was distinguishable by the highest mean number of non-roommates and highest frequency of interacting with non-roommates in the common area. Twenty-nine per cent of the participants belonged to Cluster 4: Roommate-focused (bedroom) social cluster. This cluster was characterised by a relatively substantial mean number of roommates as network partners, combined with a relatively low frequency of interaction with non-roommates. Finally, 17% of the participants belonged to Cluster 5: Restricted (bedroom) social cluster which was represented by the least mean number of network partners and the least frequency of interaction with network partners.

Figure 5 is the whole social network re-organised according to the five social clusters. Residents of different bedroom types are represented by a specific shape. The figure also shows the strength ties between network members in each cluster. Although the social clusters consist of residents from various bedroom types, certain bedroom types play a significant role in certain clusters. In social cluster 1, residents of four-bedrooms (common) have a strong presence in the clusters who engage with diverse network partners in bedroom areas. Social cluster 2, where people engage with diverse network partners in the common areas, is completely represented by residents who live in four-bedrooms (corridor). Interestingly, the same group of residents also make up the largest portion of the social cluster 4 where people engage the most with non-roommates. The residents who lived in three-bedrooms appear to represent social cluster 3 where people have limited and restricted social network partners. Lastly, almost even proportions of different residents represent social cluster 5 which suggests that the social network partners of these residents tend to be roommates.

Table 6 illustrates the number of residents in each social network cluster and the bedroom type of origin, with the percentage of sample indicated in parentheses. The table suggests a tendency for residents of a certain bedroom type to have specific social network clusters. Values in bold represent the largest proportion of residents in the specific room type. Results show a small proportion (28%) of participants who lived in four-bedrooms (corridor) had a diverse social cluster in the common area. Almost half (47%) of the members in the diverse (bedroom) cluster lived in four-bedrooms (common) which implied a tendency for small-scale interaction in the bedroom areas. A substantial proportion (33%) of the members in the non-roommate-focused cluster also lived in four-bedrooms (corridor). The roommate-focused cluster was primarily made up (50%) of participants who lived in five-bedrooms. Finally, over half (57%) of the restricted group lived in three-bedrooms, which indicates that these residents have little interaction with neither roommates or non-roommates. 

### 3.4. The Associations between Bedroom Privacy and Social Life

A multiple regression was run to explain the impact of the five architectural factors on social network structure. The result is shown in Table 7. The overall bedroom privacy has a moderately significant negative correlation with residents’ degree centrality in their social networks, network size of roommates, and frequency of contact with non-roommates in the common areas. In other words, when residents have higher bedroom privacy, they are less important in day-to-day social networks and tend to have less social networks in the common areas. The variable bedroom occupancy has statistically significant negative associations with the social networks with roommates in bedrooms (*p* < 0.01); i.e., the higher the number of room occupancy, the less established social networks there are among roommates. The variable visual privacy has a significantly negative association with residents’ degree centrality, frequency of contact with roommates and non-roommates in common areas (*p* < 0.05), and network size of roommates in the common areas (*p* < 0.01). This implies that the higher control over visual privacy (being seen from common areas) results in fewer social networks in the common areas and more social ties in bedroom areas. The variable visibility has a significant positive correlation with degree centrality, number of networks with non-roommates and frequency of contact with non-roommates in both bedrooms and common areas (*p* < 0.05, *p* < 0.01). In other words, when residents have more opportunities to see what is happening in the common areas, they are likely to have more social networks with non-roommates. The variable bedroom adjacency appears to have significant negative correlations with the network size of and frequency of contact with roommates in common areas and non-roommates in bedrooms and common areas (*p* < 0.05). The variable number of transitional spaces has a statistically significant negative associations with the residents’ degree centrality, network size of non-roommates, and frequency of contact with non-roommates in bedrooms and common areas (*p* < 0.01). It also contributes to the residents’ network size of roommates and contact frequency with roommates in common areas, as well as contact frequency of non-roommates in the bedrooms (*p* < 0.05).

## 4. Discussion

The aim of this study was to investigate if and how the provision of bedroom privacy in LTC facilities might impact the social networks of residents. Our study revealed that older adults in LTC facilities have similar social network characteristics, ranging from diverse to restricted, as those who live in the community which is consistent with Wenger’s [44] qualitative work and subsequent quantitative studies [45,46]. The results supported the argument that the architectural design and provision of bedroom privacy have a moderately significant impact on the formation of social networks among residents in LTC facilities. In respect to the provision of privacy, this study has found five architectural factors to be significant predictors of residents’ social networks. The considerations for each architectural factor are discussed in the sections below.

### 4.1. Bedroom Occupancy

This study revealed that a high bedroom occupancy has a significant effect in reducing the formation of social networks among roommates. Our study shows that residents in the most crowded five-bedrooms have the weakest social networks. This aligns with precedent findings that the lack of personal control over privacy contribute to tension and conflicts between roommates [57,58,59,60]. However, a lower bedroom occupancy could also lead to a lack of social interaction, shown in the case among residents of three-bedrooms who appeared to have few substantial social networks. Therefore, having control over privacy is especially important for residents living in multi-bed rooms so that their unique privacy needs can be met for different activities ranging from having personal time, talking with roommates, or receiving personal care [10,61]. For example, movable barriers and blinds on partition walls can increase privacy when desired [10]. Floor plans that enhance individual territories in shared rooms can minimise unwanted intrusion from roommates, offering similar benefits of private rooms [7,50]. Future studies could further investigate the benefits of these types of rooms [32]. A few studies have also shown that while the initial costs of building private rooms seem higher than shared rooms, the differences could be recouped within a relatively short period of time [7,11]. The topic of construction cost requires further study in a compact urban area like Hong Kong where land comes at a high premium.

### 4.2. Visual Privacy

The results demonstrate that a lower visual privacy in the bedroom contributes to more meaningful social networks and social ties among roommates and non-roommates. The large proportion of social interaction which took place in other people’s bedrooms (as opposed to the common areas) also indicates that visual privacy is preferred when the residents socialise with others. Our study shows that being able to be seen by other residents also provides opportunities for stronger social networks among non-roommates. However, it is worthwhile to note that while the four-bedrooms facing common areas appeared to have a large social network size, the social ties are not as strong among the residents in the four-bedrooms facing the corridor who had visibly fewer but stronger social ties. Having appropriate proportions of open versus solid surfaces on walls or partitions can prevent visual and acoustic overstimulation which maintains residents’ privacy while enabling a sense of connection with neighbours [10,50]. Being supervised by staff also promotes a sense of security and safety [10,23,33] and alleviates anxiety among staff [29,33]. It was found that, among residents who are older, they preferred lower partition walls (1.85 m) so that they can be seen should an accident take place [10]. While being supervised by the staff is not all bad, it is worth noting that residents should not be limited to spending time at locations that are visible to the staff [50]. The findings reinforce the importance of achieving a balance between visual privacy and positive social connections. Balancing safety with autonomy in a person-centred manner is a delicate balance between supporting remaining independence and choices for the individual, while recognising that sometimes systems need to be in place to mitigate risks for individuals living with dementia [50].

### 4.3. Visibility

The results showed that a lower visibility to the common areas is associated to decreased social networks of and social ties with non-roommates. It also lowers the importance of residents’ role in a network (degree centrality). Our study shows that although residents from four-bedrooms facing the common areas had a higher visibility, they did not have strong social ties since they also experience a lack of visual privacy. The findings reinforce findings from previous studies that visibility is closely linked with the perception of privacy and social networks. An increasing number of studies have adopted visibility graph analysis and isovist analysis to quantify the influence of visibility on social networks [2,3]. These studies consistently state that the building configuration of LTC facilities should be considered to meeting the social needs of older adults. Traditional nursing home design typically comprised shared bedrooms (usually 2–4 residents per room) arrayed along a long, straight corridor. Letter plans (e.g., T, H, L) were common, meaning that shared social space location was often not directly visible to residents from their bedrooms [62]. Being able to see a social space from the bedroom helps the residents with their decision-making process about being there [62]. Having a suitable open surface area, such as windows to outside views, will provide residents with interesting things to look at and talk about [10,29,63]. A radial building configuration where common areas such as lounge, dining room, and kitchen are visually accessible from the bedrooms encourages participation from the residents [64,65]. A small-scale home-like environment has been found to be especially favourable for facilitating social gatherings [32,66].

### 4.4. Bedroom Adjacency

Our study revealed that having a semi-private or semi-public space (transitional spaces) near the bedroom has a notable influence on residents’ social networks, especially with non-roommates. This reinforces the findings from previous studies on the importance of proximity between the bedroom and common areas. Studies found that residents spent more time in living and dining rooms which are visible from and near the bedrooms [28,32]. It is generally agreed that by reducing the size of LTC facilities, a home-like layout can be more easily achieved [66]. By dividing a large LTC facility (30 or more residents) into multiple smaller units with closer proximity between resident rooms and nursing stations, personal care could be enhanced which will potentially alleviate the behavioural and psychosocial symptoms of dementia [67]. According to nurses’ perspectives, proximity can be enhanced by considering the types of walls and distance between the room and nursing stations [33]. For Koncelik [27], having a small patio in front of each bedroom would give residents the capability to invite others over, turning a sterile long corridor into a “neighbourhood”. While previous studies report mixed results about residents’ agitation and behavioural problems between large and small units, they consistently find that residents of smaller LTC facilities experienced increased social engagement and higher quality of life scores [68,69].

### 4.5. Transitional Spaces

Small-scale transitional areas that support social activities such as viewing and watching are key to fostering a home-like atmosphere in LTC facilities [24,28,70,71]. The results suggest that transitional spaces contribute to degree centrality, social network size, and social ties. Our study found that the less transitional spaces between the bedroom and common areas, the more likely residents developed social networks in other bedrooms. For example, residents from four-bedrooms (common) had the least transitional spaces and spent the most time with non-roommates in their bedrooms. The lack of transitional spaces near the bedrooms has prompted the residents to utilise other people’s bedrooms as spaces for socialisation. It is also worth noting that residents from these rooms appeared to be marginalised in the overall networks and have weak social ties with other residents (Figure 4). This aligns with the claim by Calkins [50] that every environment should, in theory, offer a wide range of places, from private to semi-private to semi-public to public, so that people can decide where, when, and how they want to spend their time [25,26]. In a crowded LTC facility, seatings for one or two people along the corridor can become the hub for informal social engagement [1]. Transitional spaces are especially important for supporting the social activities of watching and observing which are essential in LTC facilities [70,71]. In addition, the provision of small-scale seating arrangements in the common areas will allow more privacy in the public space which will facilitate social interaction and the formation of social networks [1].

## 5. Conclusions

This study has shown the application of an integrated spatio-social network analysis approach as methodological and epistemological approaches that can support both quantitative and qualitative research [15]. A physical environment that has a gradation of primary, secondary, and public territories will establish a sense of privacy [9]. However, the needs for privacy in LTC facilities are also embedded in culture. The design of the territories should therefore consider the unique social needs of residents and enable opportunities for cultivating social networks. 

Bedroom occupancy should be lower whenever possible. However, this study found that residents with the most privacy were also at a higher risk of restricted networks. These bedrooms might consider wall openings through which residents can see or communicate with neighbours and staff [10]. For visual privacy, installations such as movable screens should be provided to minimise visual intrusion. Our study suggests that being seen is an important factor to social networks. Therefore, these installations should be easily adjustable so that the residents can decide when to be seen. Residents should have clear visibility of the common areas from their bedrooms as results indicate that better visibility is associated with substantially more social networks. 

Bedrooms should be within close proximity to the common areas. Our study shows that when bedrooms are organised in a small cluster, e.g., the four-bedrooms along the corridor, residents tended to have a non-roommate-oriented social network type. These results suggest that residents may prefer to interact in semi-private spaces close to their bedrooms. Transitional spaces should be planned according to the territories from the bedrooms. There should be diverse transitional spaces, ranging from nooks and niches to small-scale seating arrangement, to encourage social activities of viewing and watching. 

Recognising the unique individual social needs of residents raises questions for future research on spatial interventions to optimise the cultivation of social networks that promote residents’ psychosocial well-being. In compact shared bedrooms, residents should have access to customised privacy control mechanisms in the primary territory, more choices of semi-private spaces in the secondary territory, and small-scale seating arrangements in the public territory that is near the bedroom.

Our study could provide substantial evidence to support the amendments of the Residential Care Home Legislation Bill 2022 to increase the minimum floor area per resident and to enhance residents’ dignity and privacy for long-term care facilities in Hong Kong [72,73]. The findings of this study could significantly support the evaluation of future facilities in the privacy category. Furthermore, the findings of this study could benefit the design and development of LTC facilities in Hong Kong by advocating communications between the experts in nursing care and environmental gerontology. Moreover, the findings of this study could provide useful insights for compact LTC facilities in other high-density cities worldwide.

## 6. Future Areas of Research

This study examined five architectural factors which could influence privacy and social networks of residents. However, there were a number of key limitations in this study that could affect the findings. First, other environmental factors might also affect privacy such as unit size, acoustic control, and spatial configuration. Future study could investigate focusing on quantifying these factors according to unified and reliable standards, making results comparable across different studies. Second, we were not able to assess the effects of individual characteristics, such as socioeconomic status and health conditions on the preference of privacy and social network partners. Further examination in this perspective is needed. Third, another limitation was that only one LTC facility was investigated. Future study could examine different bedroom layouts in multiple LTC facilities so that more thorough design considerations may be created. Fourth, this study did not track the locations of every resident and staff and may have missed some interactions which might misrepresent the residents’ social networks, especially those in the restricted group (i.e., three-bedroom users). Future study could include the perspectives of staff and residents according to their design preferences and lived experiences through participatory engagement methods. Finally, the data sample of 48 participants may have been too small to generalise the influence of bedroom privacy on residents’ social networks. It is recommended that future research could include more participants from different LTC facilities.

## Figures and Tables

**Figure 1 ijerph-20-05494-f001:**
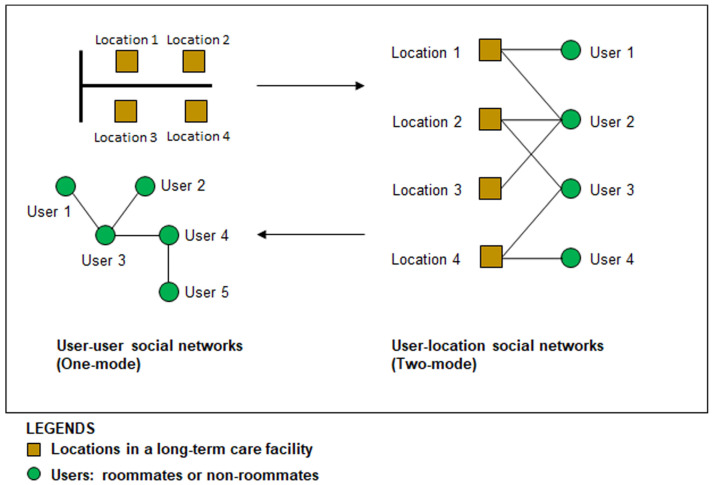
The concept of converting two-mode to one-mode social networks.

**Figure 2 ijerph-20-05494-f002:**
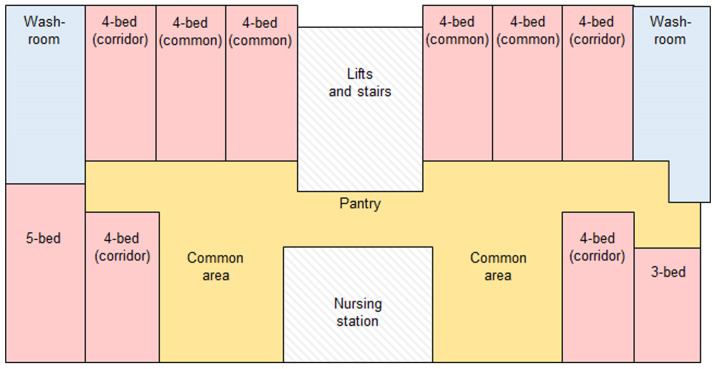
Typical floor plan of study site.

**Figure 3 ijerph-20-05494-f003:**
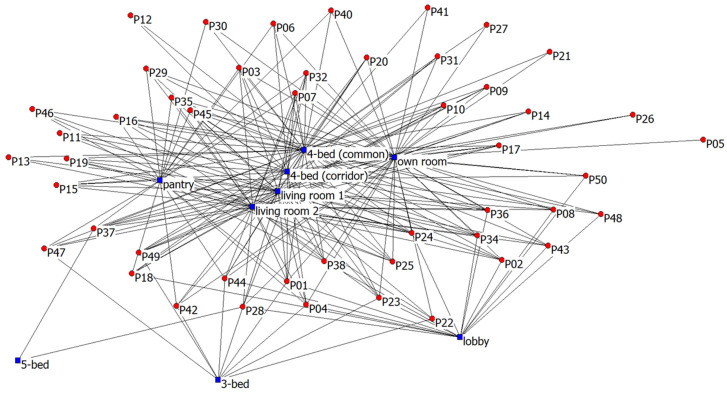
User-location (two-mode) social network analysis.

**Figure 4 ijerph-20-05494-f004:**
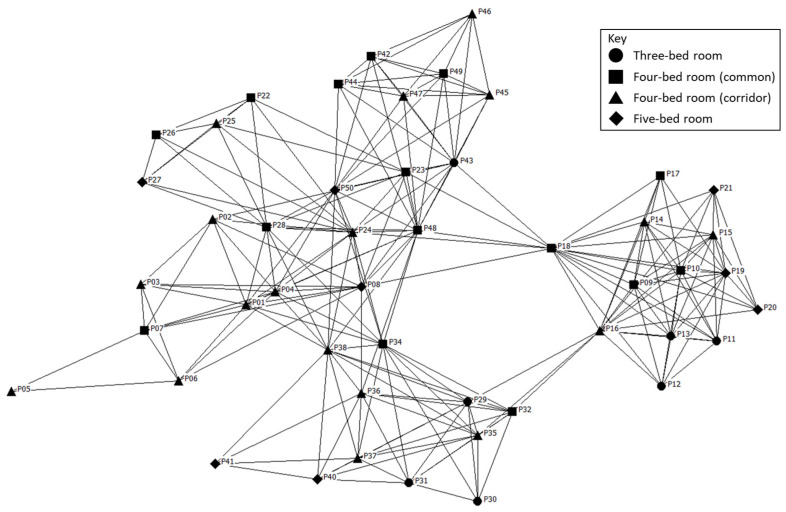
Day-to-day social networks between roommates and non-roommates.

**Figure 5 ijerph-20-05494-f005:**
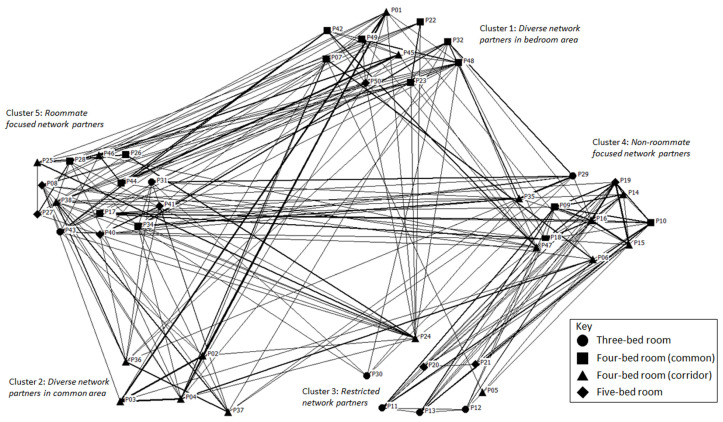
Social network re-organised according to social clusters.

**Table 1 ijerph-20-05494-t001:** Assessment of bedroom privacy in shared bedrooms.

Factor	Geometrical Indicators	References
Bedroom occupancy	How many people share the bedroom? (Five or more = 0; Four = 1; Three = 2; Two or less = 3)	[24,50]
Bedroom adjacency	What type of space is the bedroom next to? (Public space = 0; Semi-public space = 1; Semi-private space = 2; Private space = 3)	[26,27]
Transitional spaces	How many transitional spaces between the bedroom and common area? (None = 0; One = 1; Two = 2; Three or more = 3)	[1,28,29]
Visibility	Can residents see the common areas? (Yes = 0; No = 1)	[30,31]
Visual privacy	Can residents’ beds be seen from the common areas? (Yes = 0; No = 1)	[4,32]

**Table 2 ijerph-20-05494-t002:** Participant characteristics.

	Three-Bedroom	Four-Bedroom (Corridor)	Four-Bedroom (Common)	Five-Bedroom	Total
% of sample	14.58%	37.50%	31.25%	16.66%	100%
Gender					
Male	1	11	9	3	24
Female	6	7	6	5	24
Total	7	18	15	8	48

**Table 3 ijerph-20-05494-t003:** Privacy score for each bedroom type.

Architectural Factor	Three-Bedroom	Four-Bedroom (Corridor)	Four-Bedroom (Common)	Five-Bedroom
Bedroom occupancy	2	1	1	0
Bedroom adjacency	2	1	0	1
Transitional spaces	2	1	0	2
Visual privacy	1	1	0	1
Visibility	0	1	1	0
Total	7	5	2	4

**Table 4 ijerph-20-05494-t004:** Social network characteristics.

Residents of	Number of Network Partners	Frequency of Interaction	Degree Centrality
Three-bedroom	1.10	2.86	0.07
Four-bedroom (common)	2.43	4.60	0.04
Four-bedroom (corridor)	1.85	4.44	0.23
Five-bedroom	2.79	2.75	0.06
Mean	2.08	3.98	0.12

**Table 5 ijerph-20-05494-t005:** Cluster Analysis Results.

Cluster	Cluster 1. Diverse (Common Area)	Cluster 2. Diverse (Bedroom)	Cluster 3. Non-Roommate-Focused (Common Area)	Cluster 4. Roommate-Focused (Bedroom)	Cluster 5. Restricted (Bedroom)	Mean
Of sample	10.42%	20.83%	22.92%	29.17%	16.67%	
Gender						
Male	90.00%	60.00%	18.18%	57.14%	25.00%	
Female	10.00%	40.00%	81.82%	42.86%	75.00%	
Mean number of network partners by location						
Roommates in own bedrooms	2.33 (0.21)	**2.93 (0.48)**	**1.36 (0.77)**	**2.59 (0.65)**	**0.95 (0.44)**	2.03
Non-roommates in other bedrooms	**4.80 (1.48)**	4.50 (1.72)	4.27 (1.56)	3.71 (0.10)	**2.88 (1.96)**	4.03
Roommates in common areas	**0.68 (0.29)**	0.08 (0.07)	0.07 (0.09)	0.05 (0.10)	**0.01 (0.01)**	0.18
Non-roommates in common areas	1.09 (0.47)	1.21 (0.33)	**2.08 (0.61)**	**0.56 (0.41)**	**0.32 (0.34)**	1.05
Mean frequency of interacting with …						
Roommates in own bedrooms	18.60 (1.67)	**22.60 (2.46)**	**11.18 (6.06)**	**20.00 (2.86)**	**8.75 (2.76)**	16.23
Non-roommates in other bedrooms	22.80 (9.78)	**27.80 (6.73)**	24.36 (9.56)	17.36 (6.22)	**8.13 (5.91)**	20.09
None, being away from others	2.00 (3.94)	**2.50 (3.37)**	1.55 (1.57)	0.50 (0.94)	**0.13 (0.35)**	1.33
Roommates in common areas	**11.60 (3.65)**	1.50 (1.78)	4.09 (3.33)	**1.00 (1.52)**	**0.63 (0.74)**	3.76
Non-roommates in common areas	**13.40 (4.04)**	11.20 (2.78)	**14.82 (4.29)**	**5.36 (3.34)**	**2.13 (2.70)**	9.38

Note: Means with standard deviations in parentheses. Values in bold are 0.5 standard deviation above or below the mean.

**Table 6 ijerph-20-05494-t006:** Social cluster types among residents of different bedrooms.

	Bedroom	Three-Bedroom	Four-Bedroom (Corridor)	Four-Bedroom (Common)	Five-Bedroom
Cluster Type					
Diverse (in common area)		0 (0.0%)	**5 (27.8%)**	0 (0.0%)	0 (0.0%)
Diverse (in bedroom)		0 (0.0%)	2 (11.1%)	**7 (46.7%)**	1 (12.5%)
Non-roommate-focused (in common area)		1 (14.3%)	**6 (33.3%)**	3 (20.0%)	1 (12.5%)
Roommate-focused (in bedroom)		2 (28.6%)	3 (16.7%)	5 (33.3%)	**4 (50.0%)**
Restricted (in bedroom)		**4 (57.1%)**	2 (11.1%)	0 (0.0%)	2 (25.0%)
Total		7	18	15	8

Note: Number of residents with percentage in parenthesis. Values in bold represent the largest proportion of residents from a specific room type.

**Table 7 ijerph-20-05494-t007:** Associations between architectural factors of privacy and social networks.

Dependent Variables	Overall Privacy	Bedroom Occupancy	Visual Privacy	Visibility	Bedroom Adjacency	Transitional Spaces
Degree centrality	−0.305 * (0.035)	0.055 (0.709)	−0.343 * (0.017)	0.381 ** (0.008)	−0.349 * (0.015)	−0.414 ** (0.003)
Number of network partners according to location						
Roommates in bedrooms	−0.087 (0.558)	−0.498 ** (<0.001)	0.191 (0.193)	0.056 (0.704)	−0.087 (0.555)	0.08 (0.587)
Non-roommates in bedrooms	−0.225 (0.125)	0.045 (0.759)	−0.222 (0.13)	0.489 ** (<0.001)	−0.304 * (0.036)	−0.404 ** (0.004)
Roommates in common areas	−0.321 * (0.026)	0.027 (0.856)	−0.380 ** (0.008)	0.169 (0.251)	−0.315 * (0.029)	−0.317 * (0.028)
Non-roommates in common areas	−0.173 (0.241)	0.035 (0.812)	−0.188 (0.201)	0.266 (0.067)	−0.209 (0.155)	−0.259 (0.075)
Frequency of contact with …						
Roommates in bedrooms	0.091 (0.54)	−0.132 (0.372)	0.203 (0.167)	0.102 (0.489)	0.055 (0.711)	0.061 (0.679)
Non-roommates in bedrooms	−0.252 (0.084)	−0.118 (0.423)	−0.182 (0.217)	0.358 * (0.012)	−0.298 * (0.04)	−0.307 * (0.034)
Roommates in common areas	−0.241 (0.098)	0.151 (0.304)	−0.337 * (0.019)	0.274 (0.059)	−0.27 (0.064)	−0.351 * (0.014)
Non-roommates in common areas	−0.321 * (0.026)	0.033 (0.826)	−0.345 * (0.016)	0.402 ** (0.005)	−0.367 * (0.01)	−0.428 ** (0.002)

Note: Correlation coefficients are shown, with *p*-values in parentheses. * Correlation is significant at the 0.05 level. ** Correlation is significant at the 0.01 level.

## Data Availability

The data presented in this study are available on request from the corresponding author. The data are not publicly available due to its private and confidential nature.

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
