# Peer review of "Measuring the Impact of Bedroom Privacy on Social Networks in a Long-Term Care Facility for Hong Kong Older Adults: A Spatio-Social Network Analysis Approach"

_ijerph, 2023, doi:10.3390/ijerph20085494_

Round 1

Reviewer 1 Report

This article is devoted to an important issue and contains interesting findings.

However, there are some suggestions for improvement: 

Lines 173-175. <…> Social networks of the residents These elements are first calculated within the two-mode network where residents’ entire social network clusters and types are is identified according to locations.

The  part  “Social networks of the residents” looks like an incomplete sentence and should be deleted from the text.

Line 175.  <…>  and types are is identified according to locations.

The word “is” should be deleted.

Line 179.  Figure 1. From two-mode to one-mode social networks.

The figure title is incomplete.

The description of data collection and described in lines 313-324 of the Result section, subsection 3.2  “Feasibility of the spatio-social network analysis approach” could be moved to subsection 2.4 “Data collection” .

The results described in lines 405-420 would be clearer for a reader if presented in an additional table.

Lines 600-601. The findings of this study could significantly support the evaluation of future facility evaluation in the privacy category.

The second word “evaluation” in the sentence is excessive and should be removed.

Author Response

Dear Reviewer 1,

Thank you for the comments and suggestions. Kindly see our responses in the attachment. Thank you.

Reviewer 2 Report

The aim of the present study assessed the impact of bedroom privacy on residents’ social networks in a long-term care (LTC) facility for older adults. The authors bring up an interesting topic. The manuscript is well written in an engaging and lively style. 

Point 1: Does the introduction provide sufficient background and include all relevant references? Please describe what was lacking in the previous study and why this study was necessary.

Point 2: Are there any influences of individual characteristics such as socioeconomic status, health conditions, and social capitals on the research results? Isn't it a limitation of the study that individual characteristics are not adjusted?

Point 3: Only two limitations of the study were presented, but are there any additional ones? For example, isn't the survey of only 48 people too small to generalize? Also, a more detailed description and numbering of the strengths and limitations of the study will make it easier for readers to read.

Author Response

Dear Reviewer 2,

Thank you for the comments. Kindly see the attachment for our responses.
